# Combined Metabolomics and Transcriptomics Analysis of the Distribution of Flavonoids in the Fibrous Root and Taproot of *Polygonatum kingianum* Coll.et Hemsl

**DOI:** 10.3390/genes15070828

**Published:** 2024-06-22

**Authors:** Xinchun Mo, Ling Wang, Chenghua Yu, Can Kou

**Affiliations:** Department of Applied Technology, Lijiang Teachers College, Lijiang 674199, China; wangling1014@163.com (L.W.); 18468182962@163.com (C.Y.); koucan@126.com (C.K.)

**Keywords:** *Polygonatum kingianum* Coll.et Hemsl., medicinal plant, flavonoid biosynthesis, combined metabolomics and transcriptomics analysis, roots

## Abstract

Polygonati rhizoma, known for its distinct yellow rhizomes, is a common therapeutic and culinary plant in Far East Asia. The hue of medicinal plants is closely tied to the flavonoid biosynthesis and content levels. In this research, the fibrous root and taproot of *Polygonatum kingianum* Coll.et Hemsl. were studied to explore the secondary metabolite expression and flavonoid biosynthesis mechanisms using transcriptomics and metabolomics. Metabolic analysis identified that the differentially accumulated metabolites (DAMs) in the fibrous root and taproot were predominantly flavonoids, steroids, alkaloids, and phenolic acids. Overall, 200 flavonoids were identified in *P. kingianum* Coll.et Hemsl., with 170 exhibiting variances between the fibrous root and taproot. The transcriptome analysis revealed that a total of 289 unigenes encoding 32 enzymes were annotated into four flavonoid biosynthesis pathways, which include phenylpropanoid biosynthesis pathway, flavonoid biosynthesis pathway, isoflavonoid biosynthesis pathway, and flavone and flavonol biosynthesis pathway. The integration of transcriptomic and metabolomic data elucidated that the 76 differentially expressed genes (DEGs) encoding 13 enzyme genes (*HCT*, *CCOMT*, *C4H*, *C3′H*, *CHI*, *PGT1*, *FLS*, *F3′H*, *CHS*, *ANR*, *DFR*, *F3′5′H*, and *LAR*) and 15 DAMs preferred to be regulated in the flavonoid biosynthesis pathway. The expression of 10 DEGs was validated by qRT-PCR, agreeing with the same results by RNA-Seq. These findings shed light into the biosynthesis of secondary metabolites in *P. kingianum* Coll.et Hemsl., offering valuable information for the sustainable utilization and enhancement of this plant species.

## 1. Introduction

*Polygonatum* rhizome, a perennial herbaceous plant in the Liliaceae family, is primarily found in China, North Korea, Mongolia, eastern Siberia, and Russia. Historically, it has been used in therapeutics and gastronomy [1]. In China, the dried rhizomes of *Polygonatum cyrtonema* Hua, *Polygonatum kingianum* Coll.et Hemsl., and *Polygonatum sibiricum* are commonly employed as a traditional herb by local people [2]. *Polygonatum*, predominantly cultivated in southwest China, is known for its high content of polysaccharides, saponins, flavonoids, and other bioactive components, which show high-performance anti-tumor, antioxidant, anti-aging, and anti-diabetic properties [3,4,5,6]. Also, its rhizomes are a source of nutrients like starch, polysaccharides, proteins, amino acids, and vitamins, making them popular in food processing, stews, and health products. This versatile plant variety holds significant market potential and development value [7].

*Polygonatum* taproots are primarily used for medicinal purposes, with the fibrous roots constituting approximately 15% of the total rhizomes. These roots are often discarded or incinerated during extraction and processing, leading to missed opportunities for maximizing profits and posing environmental pollution and forest fire risks [8,9]. Recent studies have focused on analyzing the chemical compounds in the fibrous roots of various botanical medicinal plants like *Panax notoginseng* [10], ginseng [11,12], red ginseng [13], *Solanum dulcamara* [14], among others. This has broadened the scope of the applications for botanical medicine using fibrous roots in fields such as food and health products. Research indicates that *Polygonatum* fibrous roots contain polysaccharides, flavonoids, saponins, amino acids, and other active substances, with promising development potential and application value [15,16].

Flavonoids, commonly found in natural plants, are a series of active compounds showing various pharmacological effects, including hypoglycemic, anti-tumor, and anti-inflammatory properties [17]. They serve as key components in numerous botanical drugs. *Polygonatum*, in particular, is abundant in bioactive flavonoids like chalcones, dihydroflavones, and isoflavones, which are essential for determining the quality of *Polygonatum* [18]. The biosynthesis of flavonoids is regulated by a series of enzymes such as phenylalanine ammonia lyase (PAL), cinnamic acid 4-hydroxylase (C4H), chalcone synthase, chalcone isomerase (CHI), flavanone 3-hydroxylase (F3H), flavonoid 3′,5′-hydroxylase (F3′5′H), flavonol synthase (FLS), and dihydroflavonol 4-reductase (DFR), as well as transcription factors like MYB, bZIP, and bHLH [19]. These enzymes and factors are unequally distributed across different parts of the botanicals, leading to varying concentrations of flavonoids in different plant components [20,21,22]. Therefore, a comparative analysis of the biosynthesis and distribution of flavonoids in the taproot and fibrous root of *P. kingianum* Coll.et Hemsl. holds significant importance for its practical applications and the development of improved varieties.

The application of combined transcriptomics and metabolomics analysis has arisen as a valuable approach for exploring the biosynthesis and regulatory mechanisms of bioactive compounds in medicinal plants [23,24]. In this study, we employ metabolomics and transcriptomics to examine the variations in metabolites and gene expression levels between the taproot and fibrous root of *P. kingianum* Coll.et Hemsl., shedding light on the distribution of flavonoids and their associated regulatory genes in these root types. These findings offer a scientific foundation for the rational utilization and advancement of *P. kingianum* Coll.et Hemsl.

## 2. Materials and Methods

### 2.1. Sample Collection

*P. kingianum* Coll.et Hemsl. was planted in Shiping county, Honghe Hani and Yi autonomous Prefecture, Yunnan Province, China (102.43° E, 23.96° N), in 2019. Three 4-year-old plants with similar growth conditions were collected in June 2023. Healthy and uniform rhizomes from each plant were obtained from the whole plant. Then, the taproots (WRs) and fibrous roots (RRs) of each plant were carefully separated and immediately frozen in liquid nitrogen to preserve the RNA integrity for the transcriptome analysis. Also, the frozen WR and RR samples were used to extract the metabolites for metabolite detection. Each of the WR and RR samples from the different plants were sequenced and the metabolites were detected a total of three times. All of the samples were stored at −80 °C for the metabolomics and transcriptomics analysis and the quantitative real-time PCR (qRT-PCR) analysis.

### 2.2. Metabolites Extraction from WR and RR of P. kingianum Coll.et Hemsl.

The WR and RR samples were lyophilized using a vacuum freeze-dryer (model: Scientz-100F, Ningbo Scientz Biotechnology Co., Ltd., Ningbo, China) and then crushed with a mixer mill (model: MM 400, Verder Shanghai Instruments and Equipment Co., Ltd., Shanghai, China) through zirconia beads for 1.5 min at 30 Hz. An amount of 50 mg of lyophilized powder was dissolved with 1.2 mL of a 70% methanol solution; then, it was vortexed for 30 s every 30 min, which was repeated six times. Prior to the ultra-performance liquid chromatography–tandem mass spectrometry (UPLC-MS) analysis, we filtrated the extracts by using SCAA-104 (0.22 μm pore size; ANPEL, Shanghai, China, http://www.anpel.com.cn/ (accessed on 17 July 2023))after centrifugation at 12,000 rpm for 3 min [25].

### 2.3. Metabolome Analysis by UPLC-ESI-MS/MS

We examined the sample extracts with a UPLC-ESI-MS/MS system (UPLC, ExionLC™AD, https://sciex.com.cn/ (accessed on 17 July 2023)) and a tandem mass spectrometry system (https://sciex.com.cn/ (accessed on 17 July 2023)) with the following conditions: UPLC column, Agilent SB-C18 (1.8 µm, 2.1 mm × 100 mm); the mobile phase consisted of solvent A, which was composed of pure water with 0.1% formic acid, and solvent B, which was composed of acetonitrile with 0.1% formic acid. Sample measurements were conducted using a gradient program applying the preliminary solution consisted of 95% solvent A and 5% solvent B and maintaining for 9 min, a linear gradient to 5% solvent A and 95% solvent B was automated and reserved for 1 min. Next, we modified the proportions to 95% solvent A and 5% solvent B within 1.1 min, kept for 2.9 min. Then adjusted the flow velocity to 0.35 mL·min^−1^, the column oven to 40 °C, and the injection volume to 2 μL. Conversely, we linked the effluent to an ESI-triple quadrupole-linear ion trap (QTRAP)-MS.

The following were the ESI process conditions: 550 °C; an ion spray voltage (IS) of 5500 V (positive ion mode)/−4500 V (negative ion mode); ion source gas I (GSI), gas II(GSII), and curtain gas (CUR) set at 50, 60, and 25 psi, respectively; high collision-activated dissociation (CAD). The QQQ scans were developed as multiple reaction monitoring (MRM) experiments with the collision gas (nitrogen) set to medium. The DP (declustering potential) and CE (collision energy) for the individual MRM transitions were set with optimization parameters. An exact set of MRM transitions were conducted for each period based on the eluted metabolites within this period [25].

Unsupervised principal component analysis (PCA) was calculated by MetaboAnalystR 4.0 (https://github.com/xia-lab/MetaboAnalystR (accessed on 17 July 2023) [26]. The data included the unit variance scaled before the unsupervised PCA. The hierarchical cluster analysis (HCA) outcomes of the samples and metabolites were conducted by R package 4.3.1 and are shown as heatmaps with dendrograms. Alternatively, the Pearson correlation coefficients (PCCs) between the samples were also calculated and are shown only as heatmaps. For the HCA, the normalized signal intensities of the metabolites (unit variance scaling) were employed to visualize them with a color spectrum. For the two-group analysis, differential accumulated metabolites (DAMs) were screened and identified by the variable importance in projection (VIP) values (VIP > 1) and absolute Log_2_FC values (|Log_2_FC| ≥ 1.0), which were generated from the results of the orthogonal signal correction and partial least squares–discriminant analysis (OPLS-DA) [27], correspondingly comprising the score plots and permutation plots made by using MetaboAnalystR 4.0 [26]. The data were log-transformed (log_2_) and mean-centered before the OPLS-DA. A permutation test (200 permutations) was conducted to avoid overfitting. Identified metabolites were initially retrieved by the KEGG Compound database (http://www.kegg.jp/kegg/compound/ (accessed on 17 July 2023)) and mapped to the KEGG Pathway database (http://www.kegg.jp/kegg/pathway.html (accessed on 17 July 2023)). Significantly regulated metabolites were then appled to perform the metabolite sets enrichment analysis (MSEA) and further verified by the hypergeometric test’s *p*-values [28].

### 2.4. RNA Extraction and Transcriptome Analysis

Total RNA was isolated from the WR and RR samples by using the RNAprep Pure Plant Kit (Tiangen Biotech Co. Ltd., Beijing, China). To detect the integrity and purity, the extracted total RNA was assessed using the RNA Nano 6000 Assay Kit of the Bioanalyzer 2100 system (Agilent Technologies, Santa Clara, CA, USA) and the NanoPhotometer^®^ spectrophotometer (IMPLEN, Westlake Village, CA, USA), respectively. The RNA concentration was evaluated with the Qubit^®^ RNA Assay Kit in the Qubit^®^ 2.0 Flurometer (Life Technologies, Carlsbad, CA, USA). The cDNA synthesis and construction of the libraries were conducted according to the manufacturer’s instructions by using the NEBNext^®^ Ultra^TM^ RNALibrary Prep Kit for Illumina^®^ (NEB, Ipswich, MA, USA). The constructed libraries were used for the transcriptome sequencing on the Illumina NovaSeq 6000 (Illumina, San Diego, CA, USA) platform and 150 bp paired-end reads were generated.

Clean reads were generated by the raw data (raw reads), and then filtered with the FastP to remove the adapters and ploy-N and low-quality (Q < 20) reads [29]. Then, they were de novo assembled into transcripts by the Trinity 2.13.2 software [30]. The assembled transcripts were aligned to the Nr, Swiss-Prot, Tremb, KEGG, GO, KOG/COG, and Pfam databases by BLAST+ (version 2.15.0). The values of the fragments per kilobase of exon model per million mapped fragments (FPKMs) were calculated to estimate the abundance of the transcripts and gene expression levels by RSEM (version 1.3.3) [31]. Pairwise differentially expressed genes (DEGs) were screened by using DESeq2 1.22.2 software with a |log2foldchange| (FC) threshold value ≥ 2 and a false discovery rate (FDR) < 0.05 [32]. The GO classification and KEGG enrichment analysis were performed by TBtools 2.102 [33]. Transcription factor (TF) analysis was also performed by using iTAK 1.7a [34].

### 2.5. Combined Transcriptomic–Metabolomic Analysis

Based on the transcriptome and metabolome data, the DEGs and DAMs were matched with the KEGG pathway database to discover the shared pathway. By analyzing Pearson’s correlation coefficient (PCC) values > 0.8 and *p* < 0.05, the correlation between the DEGs and DAMs was investigated and visualized via a nine-quadrant plot. To gain insight into the connections between the DEGs and DAMs, network graphs were drawn with Cytoscape 3.10.2 software [35].

### 2.6. Validation of DEGs by qRT-PCR Analysis

Ten DEGs involved in the flavonoid synthesis were validated by qRT-PCR using the RT^2^ SYBR Green ROX qPCR Mastermix Kit (Qiagen, Beijing, China) and using the *ubiquitin* (*UBQ*) gene as the internal inference. Gene primers were designed and are listed in Table 1. The 10 µL reaction mixture contained 1 µL of cDNA (≤100 ng/reaction), 0.7 µL of each primer, 5 µL of the 2× SYBR qPCR Master Mix, 0.05 µL of QN ROX Reference Dye, and 2.55 µL of RNase-free water. All of the qRT-PCR analyses were performed using the following conditions: denaturation at 95 °C for 2 min, followed by 40 cycles of 95 °C for 5 s, and then at 60 °C for 30 s. All of the reactions were repeated three times in the experiments, and the 2^−ΔΔCt^ method [36] was used to calculate the relative expression of each gene.

## 3. Results

### 3.1. Multivariate Statistical Analysis

To assess the differential metabolites within the WR and RR of *P. kingianum* Coll.et Hemsl., a multivariate statistical analysis was performed (Figure 1). The PCA results showed that there were significant differences between the WR and RR samples, for which PC1 and PC2 accounted for 82.56% and 5.67% of the entire variation (Figure 1A) and formed two distinct clusters (Figure 1B). Nevertheless, the WR possessed a higher degree of separation than that of the RR within the groups. Cluster analysis also revealed that the metabolites were quite different between the WR and RR. The first group gathered more in the RR, while the second group gathered more in the WR. Furthermore, the first cluster showed a significantly smaller number of metabolites than the second cluster, suggesting significant differences in the metabolites between the WR and RR (Figure 1B).

To verify the differentially abundant metabolites between the WR and RR, OPLS-DA was employed to screen the differential metabolites within these two groups. In the OPLS-DA model, a set of prediction parameters (such as R2X, R2Y, and Q2) were predicted, where R2X and R2Y represent the interpretation rate for the X and Y matrices, and Q2 reflects the utility of the model for making predictions. Specifically, when the values of the three parameters were close to one, this suggested a reliable and stable model. The R2X value was 0.8493, the R2Y value was close to one, and the Q2 value was 0.997 (>0.9), suggesting that the OPLS-DA model could sufficiently explain and predict the differences between the two groups (Figure 1C).

The VIP values of the multivariate analysis were applied to initially screen for the differential metabolites among the two groups based on the OPLS-DA model. Furthermore, the fold-change and absolute Log_2_FC values in the univariate analysis were estimated for the further screening of differentially abundant metabolites. When the metabolites possessed VIP values ≥ one and *p*-values < 0.05, they were regarded as DAMs. A total of 1206 significant DAMs were obtained between the WR and RR, where 127 DAMs were significantly upregulated (red dots) and 1079 DAMs were significantly downregulated (green dots; Figure 1D). A total of 590 DAMs showed an insignificant difference between the two groups (gray dots). A list of the 40 most significantly upregulated and downregulated differentially abundant metabolites is provided in Table 2.

These metabolites were mostly classified into the following four groups: steroids, flavonoids, amino acids and derivatives, and alkaloids. Within the 20 upregulated metabolic components, 6 of them belonged to flavonoids, such as clitorin, kaempferol-3-*O*-rutinoside-7-*O*-rhamnoside, quercetin-7-*O*-rutinoside, trifolirhizin, quercetin-3-*O*-(2″-*O*-Rhamnosyl) rutinoside, and isorhamnetin-3-*O*-rutinoside-7-*O*-rutinoside, whereas, among the 20 downregulated metabolic components, only 2 of them belonged to flavonoids, including 4-hydroxychalcone and 4′,5,7-Trihydroxy-3′,6-Dimethoxyflavone.

### 3.2. Functional Annotation and Enrichment Analysis of DAMs

To clarify the differential metabolites within the biosynthesis pathway, the DAMs were functionally annotated by retrieving them against the KEGG database. A total of 328 DAMs were annotated into 198 metabolic pathways, including flavonoid derivatives biosynthesis pathways like phenylpropanoid biosynthesis pathway (ko00940), flavonoid biosynthesis pathway (ko00941), isoflavonoid biosynthesis pathway (ko00943), and flavone and flavonol biosynthesis pathway (ko00944). Also, some of the DAMs were detected to respond to penoid derivatives biosynthesis pathway, such as Diterpenoid biosynthesis (ko00904), Sesquiterpenoid and triterpenoid biosynthesis (ko00909), and Stilbenoid, diarylheptanoid, and gingerol biosynthesis (ko00945) pathways (Figure 2, Appendix A). Forty-four of the DAMs were assigned into the flavonoid biosynthesis (Figure 2A) and possessed a high *p*-value (Figure 2B), and only five of them were related to the penoid biosynthesis, indicating that the flavonoids were the major metabolites to be used as pharmaceutical components. Most of them (148 DAMs) were annotated to the metabolic pathways, while 71 DAMs were related to the biosynthesis of secondary metabolites. Among the pathways involved in the biosynthesis of plant secondary metabolites, more than 20 DAMs were enriched in the biosynthesis of cofactors, ABC transporters, the biosynthesis of amino acids, the nucleotide metabolism, the purine metabolism, and flavonoid biosynthesis (Appendix A).

### 3.3. Flavonoid Derivatives Identification

To understanding of the flavonoid derivatives synthesized by *P. kingianum* Coll.et Hemsl., entire metabolites were retrieved and mapped into the flavonoid pathways on the bias of the metabolomic data. The results showed that 200 flavonoid derivatives were found in the WR and RR (Appendix A), such as chalcones, flavanones, flavanonols, flavonols, flavanols, isoflavones, etc. Under the criteria of VIP value ≥ one and *p*-value < 0.05, 170 differential flavonoid derivatives were screened out in the WR and RR (Appendix A). Within these differential flavonoid derivatives, most of them (153 flavonoids) were highly synthesized in the WR, whereas only 17 flavonoids were highly synthesized in the RR (Figure 3). This suggests the reason why people usually used the WR as medicine and not the RR. Specially, 62 flavonoid derivatives were only distributed in the WR, like quercetagitrin, scillavone B diglucoside, isoquercitrin, baicalin, yuanhuanin, baimaside, and homoplantaginin. However, some of the flavonoid derivatives in the RR showed 100-fold higher contents than that of the WR, such as 4-hydroxychalcone, jaceosidin, naringenin chalcone, pinobanksin, phloretin, methylophiopogonanone B and disporopsin, naringenin, and hesperetin (Appendix A). These results suggest that the RR might be a potential medicine resource for flavonoid extract applications.

### 3.4. Functional Annotation and Enrichment Analysis of DEGs

To understanding the potential regulatory molecular mechanisms of the flavonoid biosynthesis of *P. kingianum* Coll.et Hemsl., six cDNA libraries from the WR and RR were constructed with the same samples used for the metabolite data analysis. A total of 48,027 DEGs were found, and nearly half of them showed significant regulation within the WR and RR. Of them, 9352 were significantly upregulated and 10,618 were downregulated (Figure 4). Further functional annotation towards the GO database showed that the DEGs were grouped into three categories and 46 subclasses, including 22 biological processes, two cellular components, and 22 molecular functions. Most of the DEGs were enriched in the cellular and metabolic processes of the biological processes, in the cellular anatomical entities of the cellular components, in the binding of the molecular functions, and in the catalytic activity (Figure 5A).

### 3.5. Pathway Annotation of DEGs in WR and RR of P. kingianum Coll.et Hemsl.

The results of the KEGG pathway analysis showed that 26,854 DEGs were detected and enriched in 143 KEGG pathways; of them, 6213 DEGs were related to the metabolic pathway and 289 DEGs were annotated into flavonoid-derivative synthesis pathways (Appendix A). The most significantly enriched pathways are shown in Figure 5B, including the biosynthesis of secondary metabolites, phenylpropanoid biosynthesis, metabolic pathways, flavonoid biosynthesis, the biosynthesis of various plant secondary metabolites, and zeatin biosynthesis.

The RNA-sequencing analysis results showed that a total of 289 unigenes encoding 32 enzymes were annotated into flavonoid synthesis pathways and 11 enzymes were found in *Polygonatum* (Table 3 and Appendix A). All 76 DEGs were annotated into the flavonoid biosynthesis pathway (ko00941) (Appendix A), and 28 of them were shared with the phenylpropanoid biosynthetic pathway (ko00940), while only 2 of the DEGs were associated with the flavone and flavonol biosynthesis pathway (ko00944). In the further analysis of the expression patterns of the genes regulated by metabolite biosynthesis, we found that six genes were significantly downregulated in the flavonoid biosynthesis pathway, including anthocyanidin reductase (*ANR*), trans-cinnamate 4-monooxygenase (*C4H*), 5-*O*-(4-coumaroyl)-d-quinate 3′-monooxygenase (*C3′H*), flavonoid 3′-monooxygenase (*F3′H*), flavonol synthase (*FLS*), and leucoanthocyanidin reductase (*LAR*), and metabolites like butin, dihydrofisetin, galangin, eriodictyol, kaempferol, and quercetin were also significantly decreased, indicating that those genes and metabolites of the flavonoid biosynthesis pathway were significantly decreased at this growth period. Only the 3′,5′-hydroxylase (*F3′5′H*) gene was significantly upregulated and was associated with gene *F3′H* to regulate the metabolite synthesis in the flavonoid biosynthesis pathway (Appendix A). Six genes were detected to exhibite mixed regulation, caffeoyl-CoA *O*-methyltransferase gene (*CCOMT*), chalcone isomerase gene (*CHI*), chalcone synthase gene (*CHS*), bifunctional dihydroflavonol 4-reductase gene (*DFR*), shikimate *O*-hydroxycinnamoyltransferase gene (*HCT*), and phlorizin synthase gene (*PGT1*), were led to some of the significantly downregulated metabolites, like *p*-coumaroyl quinic acid, naringenin chalcone, naringenin, and pinocembrin. Interestingly, the downregulated gene *ANR* significantly increasingly synthesized (-)-epigallocatechin from delphinidin. The results suggested that most of the genes and metabolites of the flavonoid biosynthetic pathway were significantly descended at this time, and that the ANR gene might play a potentially negative activation feedback in (-)-epigallocatechin biosynthesis (Appendix A).

### 3.6. Transcription Factor Identification in WR and RR of P. kingianum Coll.et Hemsl.

Transcription factors (TFs) have been reported to interact with numerous genes in stimulating the simultaneous expression of structural genes within secondary metabolic pathways. A total of 877 differential TFs were obtained by the transcriptome analysis from the WR and RR, which were classified into 57 transcription factor families (Appendix A). Among them, 285 TFs were upregulated and 592 were downregulated. The most abundant transcription factor families were the AP2/ERF-ERF and WRKY families, whereafter it was the C2H2, NAC, bHLH, MYB and bZIP families (Figure 6). We found three transcription factor families involved in the regulation of the biosynthesis of flavonoids, which were the MYB, bHLH, and bZIP families. Further expression level analysis revealed that the seven abundant transcription factor families had significantly expressed activity in the WR and RR samples, with 84 TFs (11 upregulated/73 downregulated) in the AP2/ERF-ERF family, 79 (10/69) in the WRKY family, 53 (11/42) in the C2H2 family, 52 (9/43) in the NAC family, 45 (10/35) in the bHLH family, 42 (9/33) in the MYB family, and 40 (12/28) in the bZIP family (Figure 6).

### 3.7. Flavonoid Biosynthesis-Related Gene Identification in WR and RR of P. kingianum Coll.et Hemsl.

On the basis of the transcriptome analysis, 76 DEGs were found to be involved in the flavonoid biosynthesis of the WR and RR. However, there was some confusion, as only 14 DEGs were shown to be upregulated, and most of them (62 DEGs) were downregulated, indicating that the period of plant growth was not the suitable time for obtaining high contents of flavonoids in *P. kingianum* Coll.et Hemsl. Seven genes were shown to be significantly downregulated in the flavonoid biosynthesis, such as Cluster-49335.3, Cluster-4499.7, Cluster-20933.0, Cluster-58756.5, Cluster-40809.0, Cluster-6224.0, and Cluster-12152.1, where Cluster-57929.0, Cluster-57929.1, Cluster-32929.0, and Cluster-64884.11 were upregulated. More than 77% (59/76) of the identified flavonoid biosynthesis-related genes were quite differently expressed within the WR and RR, which suggests that the flavonoid biosynthesis is organ-specific in its expression (Appendix A). The KOG analysis revealed that 32 gene fragments were annotated as functional genes; of them, 26 gene fragments were found to be downregulated, including 8 genes for the UDP-glucuronosyl and UDP-glucosyl transferases, 6 genes for the cytochrome P450 CYP2 subfamily members, 4 genes for flavonol reductase/cinnamoyl-CoA reductase, 3 genes for the iron/ascorbate family oxidoreductases, and 5 genes for O-methyltransferases. Six upregulated genes were also found, including UDP-glucuronosyl and UDP-glucosyl transferase, flavonol reductase/cinnamoyl-CoA reductase, and FOG: Transposon-encoded proteins with TYA, reverse transcriptase, and integrase domains in various combinations (Appendix A). Functional analysis revealed that the flavonoid biosynthesis-related genes primarily expressed in the WR were associated with flavone 7 and 3′-*O*-glycosyltransferase, UDP-glycosyltransferase, caffeine-CoA *O*-methyltransferase, trans-cinnamate 4-monooxygenase, and dihydroflavonol 4-reductase. Additionally, the anthocyanidin 3-*O*-glucosyltransferase 2- and tetraketide α-pyrone reductase-related genes were distributed in the RR. These results suggest that organ-specific genes can regulate the flavonoid biosynthesis in both the WR and RR (Appendix A).

### 3.8. Coalitional Analysis of the Transcriptome and Metabolome Analysis

To give a deep insight into the flavonoid biosynthesis mechanism in the fibrous root and taproot of *P. kingianum* Coll.et Hemsl., a comprehensive analysis combined metabolomics and transcriptomics was conducted. The results show that there is a positive correlation between the expression of genes and the flavonoid distribution in the WR and RR (Figure 7A). To calculate the relevance of the gene expression and metabolite levels, Pearson’s correlation coefficient was employed to estimate the correlation of the DEGs and DAMs involved in the flavonoid synthesis pathway, such as flavonoid biosynthesis pathway, isoflavonoid biosynthesis pathway, and flavone and flavonol biosynthesis pathway. The results revealed that 76 DEGs encoding 13 enzymes were significantly positively correlated (*p* < 0.01) with 15 DAMs in the flavonoid biosynthesis pathway, 28 DEGs with 2 DAMs in the isoflavonoid biosynthesis pathway, and 14 DEGs with 8 DAMs in the flavone and flavonol biosynthesis pathway, respectively (Figure 7B, Appendix A). It seems that these 25 metabolites prefer to be regulated in the flavonoid biosynthesis pathway, which is close to the results of the KEGG pathway analysis (Figure 2B). The results suggest that the flavonoid biosynthesis might be regulated by distinct genes in the WR and RR.

The previous results demonstrate that there are distinctly different genes in the WR and RR that regulate the flavonoid-derivative biosynthesis. Therefore, we performed a correlative analysis between the DEGs and DAMs in three flavonoid biosynthesis pathways in the WR vs. RR (Figure 8). Most of the genes were found to be significantly correlated with the flavonoid biosynthesis, isoflavonoid biosynthesis, and flavone and flavonol biosynthesis pathways. The results indicated that 61 DEGs were shown to be significantly positively correlated to those in the flavonoid-derivative biosynthesis, such as Naringenin chalcone (pme2960), Pinobanksin (mws0914), Hesperetin (mws0463), Hesperetin-7-*O*-glucoside (Lmzp002365), Catechin (mws0054), Homoeriodictyol (mws1033), Phloretin (pme1201), Quercetin (pme2954), Pinocembrin (MWSHY0124), Kaempferol (mws1068), Eriodictyol (MWSHY0145), and Naringenin (MWSHY0017). Additionally, another 13 DEGs were observed to be positively correlated with Epigallocatechin (MWSHY0098) and Naringin (mws0046) (Figure 8A, Appendix A). In the isoflavonoid biosynthesis pathway, 25 DEGs showed a positive correlation with the biosynthesis of Naringenin (MWSHY0017), while only 1 DEG was positively correlated with the biosynthesis of Trifolirhizin (mws1172) (Figure 8B, Appendix A). Within the flavone and flavonol biosynthesis pathway, 11 DEGs showed a positive correlation with the biosynthesis of 3,7-Di-*O*-methylquercetin (mws0917), Quercetin (pme2954), Baimaside (MWSHY0162), Isoquercitrin (MWSHY0046), Cynaroside (MWSHY0104), and Kaempferol (mws1068), while only 2 DEGs exhibited a positive correlation with the biosynthesis of Rutin (mws0059) and Rhoifolin (mws0047) (Figure 8C, Appendix A). These results demonstrate that the biosynthesis of flavonoids is a highly intricate biological process in *P. kingianum* Coll.et Hemsl., influenced by the coordinated regulation of multiple genes. Owing to the combination analysis of the KEGG pathway and transcriptomic and metabolomic data, we could draw a profile of the potential pathways for the flavonoid biosynthesis and accumulation in *P. kingianum* Coll.et Hemsl.

### 3.9. Validation of DEGs by qRT-PCR

To validate the accuracy of the RNA-Seq data, 10 differentially expressed genes encoding 10 key enzymes were selected for a quantitative real-time PCR (qRT-PCR) analysis. As illustrated in Figure 9, the expression patterns of these genes were consistent with the corresponding FPKM values derived from the RNA-Seq analysis (Appendix A), which indicated the reliability of the transcriptome. The nine core genes for the flavonoid biosynthesis were more highly expressed in the WR than those in the RR, except *F3′5′H*, such as *PGT1*, *CHI*, *FLS*, *ANR*, *LAR*, *CYP73A*(*C4H*), *CHS*, *DFR*, and *C3′H*.

## 4. Discussion

*Polygonatum*, a widely recognized medicinal and edible plant, has been demonstrated to possess various biological activities and a potentially significant market status [41,42].

Metabolomic research revealed distinct differences in the metabolites between the WR and RR of *P. kingianum* Coll.et Hemsl., highlighting significant variations in the secondary metabolite composition (Figure 1). Secondary metabolites, which serve as the active constituents of medicinal plants, are important in the growth, development, reproduction, and other life processes of plants [43,44,45]. Metabolome analysis showed that the major constituents of the secondary metabolites in *P. kingianum* Coll.et Hemsl. are predominantly found in the taproot, such as flavonoids, phenolic acids, alkaloids, steroids, and terpenes. The taproot, being the primary medicinal and edible part of *P. kingianum* Coll.et Hemsl., is abundant with these secondary metabolites, which form the foundation of its high medicinal value [46,47]. Steroids and flavonoids, the key secondary metabolites in plants, exhibit diverse pharmacological effects, such as antioxidant, anti-inflammatory, anti-tumor, antibacterial, antiviral, and analgesic properties [48,49,50,51,52]. Despite the detection of only 142 secondary metabolites in the fibrous roots of *P. kingianum* Coll.et Hemsl., they are notably rich in steroids and flavonoids, providing a basis for the appropriate utilization of the fibrous roots.

Flavonoids, including flavonols, flavonoids, chalcones, and proanthocyanidins, are natural active substances with significant health benefits and potential applications in healthcare and medicine [53,54]. In this study, we conducted a metabolomics analysis on the WR and RR of *P. kingianum* Coll.et Hemsl., and 200 flavonoid compounds were identified from these two organs. The taproot contained a higher quantity of flavonoids compared to the fibrous roots, where the taproot contained 153 flavonoids such as 4-Hydroxychalcone, Naringenin chalcone, Pinobanksin, Naringenin, Hesperetin, and Kaempferol, and the fibrous roots were comprised of 17 flavonoids, including kaempferol, apigenin, isorhamnetin, quercetin, and glycoside metabolites of rhamnetin. Interestingly, flavonoids in the fibrous roots also exhibit diverse activities, offering a material basis for their potential applications in food and medicine. For instance, quercetin shows strong anti-diabetic properties and can reduce diabetic renal tissue damage, renal oxidative stress, and inflammation in streptozotocin-induced diabetic rats [55]. Kaempferol has demonstrated antiviral effects, the regulation of allergic airway inflammation and asthma, as well as the prevention of gastric inflammation and alcohol-induced gastric damage [56,57,58]. Additionally, naringenin and naringin exhibit neuroprotective and antibacterial activities [59,60].

Currently, there is still a lack of reports on the enzyme genes associated with flavonoid production in *P. kingianum* Coll.et Hemsl. Here, we identified a total of 76 expressed genes involved in flavonoid synthesis, with 59 genes showing differential expression between the taproot and fibrous roots. Specifically, 23 genes significantly upregulated in the taproot were found to be associated with three types of enzymes involved in catalyzing flavonoid synthesis. These findings suggest that the levels of flavonoids in the WR and RR of *P. kingianum* Coll.et Hemsl. are influenced by the expression of genes related to flavonoid production.

The biosynthesis of secondary metabolites is intricately linked to the expression of structural genes and transcription factors [61]. Transcription factors play a crucial role in regulating the active expression of synthetases, with the transcriptional activation of synthetic factors being a key regulatory mechanism in the plant secondary metabolism [62,63]. Numerous studies have identified various families of transcription factors (MYB, MYC(bHLH), bZIP, and WD40) involved in the biosynthesis of flavonoids in medicinal plants like dendrobium [24], astragalus [64], sea buckthorn [65], and jujube [66]. In this study, it was observed that transcription factors such as MYB, bHLH, and bZIP, known to regulate flavonoid biosynthesis, were also present in the WR and RR of *P. kingianum* Coll.et Hemsl. The differential expression of these transcription factors in the WR and RR of *P. kingianum* Coll.et Hemsl. holds significant implications for understanding the biosynthesis mechanisms and differential distribution of flavonoids in these root structures.

Joint analysis of the transcriptome and metabolome can be instrumental in identifying the biosynthetic pathways of pharmacologically significant metabolites. Through a combined correlation analysis of DAMs and DEGs in the WR and RR of *P. kingianum* Coll.et Hemsl., key genes and pathways involved in flavonoid synthesis were pinpointed. Specifically, in the flavonoid biosynthesis pathway, 12 DAMs were found to be positively correlated with 61 DEGs; in the isoflavonoid biosynthesis pathway, 2 DAMs showed a positive correlation with 25 DEGs; and in flavone and flavonol biosynthesis pathway, 6 DAMs exhibited a positive correlation with 11 DEGs, highlighting the distinct relationships between gene expression and metabolites. Furthermore, a comprehensive analysis of the genes regulating metabolite synthesis through transcriptional and metabolomic approaches can offer insights into the molecular-level elucidation of the biosynthetic pathways of secondary metabolites.

The combined analysis of transcriptomics and metabolomics revealed that 44 DAMs were annotated into the flavonoid and related derivative biosynthetic pathway, which significantly associated with 76 DEGs encoding 32 enzymes (Figure 2A, Table 2). De novo transcriptome assembly and metabolomic analysis revealed that 76 DEGs (*HCT*, *CCOMT*, *C4H*, *C3′H*, etc.) and 200 metabolites (Naringenin chalcone, Pinobanksin, Naringenin, etc.) are involved in the flavonoid biosynthesis pathways and related networks (Appendix A, Table 3) [55]. The combined transcriptome and metabolome analysis of the roots in *P. kingianum* Coll.et Hemsl. revealed that the expression patterns in differential genes and flavonoid derivatives were significantly related [67]. Among the DEGs, 13 genes (*ANR*, *CCOMT*, *C4H*, *C3′H*, *CHI*, *CHS*, *DFR*, *F3′H*, *F3′5′H*, *FLS*, *HCT*, *LAR*, and *PGT1*) related to flavonoid biosynthesis also showed significant correlations with the DAMs (Appendix A). The CHS enzyme was reported to be one of the most explored enzymes in medicinal plants, and was the major rate-limiting enzyme in the flavonoid biosynthesis pathway, which catalyzed to form chalcone, a substance to further form other different flavonoids [37,38,39,68]. Some flavonoid derivatives, such as phloretin and naringenin chalcone generated from p-coumaroyl-CoA content, were negatively correlated with the expression of the *CHS* gene in the WR and RR of *P. kingianum* Coll.et Hemsl.; the same trend was also observed during the fruit development of *Lycium chinense* [69]. This might have contributed to the differential spatiotemporal expression or functional redundancy of the *CHS* genes in the WR and RR of *P. kingianum* Coll.et Hemsl. The *CHI* gene stereospecificially cyclizated the chalcone (e.g., naringenin chalcone) to produce the corresponding flavanone (e.g., naringenin), which was then partitioned into the biosynthetic pathways of the flavones, flavonols, anthocyanins, proanthocyanidins, and other derivatives [70]. The expression of the *DFR* gene was highly positively correlated with the content of seven flavonoid derivatives in this study, such as *O*-5-deoxyle ucopelargonidin, *O*-5-deoxyle ucocyanidin, leucopelargonidin, leucocyanidin, apiforol, luteoforol, and leucodelphinidin (Appendix A). It was reported to be an essential enzyme responsible for the diversification of anthocyanins and ellagitannins [71], and positively correlated with the accumulation of flavonoids against heavy metal stress in leaves of field conifers [72]. In the red cell line of *Saussurea medusa*, the *DFR* gene was found to show a significantly positive response to the accumulation of catechin and the synthesis of eriodictyol [73]; however, we found that another function of the *DFR* gene was the accumulation of apiforol and luteoforol from naringenin and eriodictyol (Appendix A). The *LAR* gene was detected to be significantly downregulated by the accumulation of (+)-catechin, but not influenced by (+)-afzelechin and (+)-gallocatechin (Appendix A), where it was reported as a key enzyme in catalyzing the conversion of colorless anthocyanins into catechins in the flavonoid synthesis pathway [74].

The majority of genes involved in the flavonoid biosynthesis pathway were found to exhibit heightened expression levels in the WR and showed lower expression levels in the RR, which was further validated by qRT-PCR (Figure 3 and Figure 9). As a result, the most differential metabolites were also identified between the WR and RR (Figure 3). These preliminary results shed light on the relevance of the metabolites and genes in the flavonoid synthesis pathway, laying the foundation for further studies of the specific mechanisms of the secondary metabolite biosynthesis in the roots of *P. kingianum* Coll.et Hemsl.

## 5. Conclusions

Metabolomic analysis revealed 1,206 metabolic differential compounds in the taproots and fibrous roots of *P. kingianum* Coll.et Hemsl., with the fibrous roots containing 142 metabolites with a higher content. A total of 200 types of flavonoids were identified in both root types, with 170 showing a distinct distribution in the taproots, such as flavones, flavonols, dihydroflavones, chalcones, and isoflavones, among which 17 types were notably abundant in the fibrous roots. The transcriptomic analysis revealed that 76 genes associated with flavonoid synthesis were identified in the fibrous roots and taproots. Transcription factors like MYB, bHLH, and bZIP were found to regulate flavonoid synthesis. By combining transcriptomics and metabolomics, key genes and pathways involved in the flavonoid synthesis in the taproots and fibrous roots of *P. kingianum* Coll.et Hemsl. were identified. Differentially expressed genes (DEGs) related to flavones, isoflavones, flavonoids, and flavonols showed significant correlations with differentially accumulated metabolites (DAMs). However, further research is required to clarify the precise functions of particular genes and transcription factors in the flavonoid biosynthesis pathway. These results contribute to understanding the potential mechanisms underlying variations in the flavonoid biosynthesis and distribution in roots, providing reference for the further application and development of the taproots and fibrous roots of *P. kingianum* Coll.et Hemsl.

## Figures and Tables

**Figure 1 genes-15-00828-f001:**
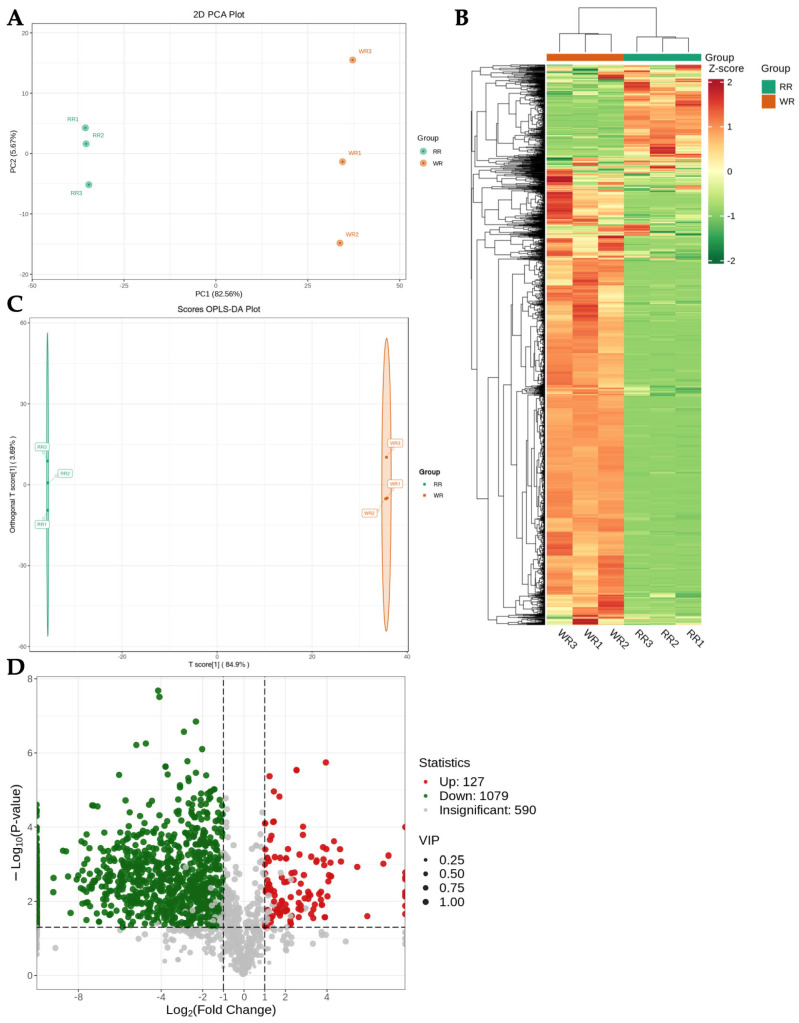
Multivariate statistical analysis in the WR and RR of *P. kingianum* Coll.et Hemsl. (**A**). PCA analysis on WR and RR group. (**B**). Cluster heatmap of the WR and RR samples and metabolite classification. The cluster lines on the left represent the metabolite clusters. The cluster lines at the top represent sample clusters. Red and green indicate high and low metabolite contents, respectively. (**C**). PLS-DA score plots of WR and RR. (**D**). Volcano plot of differential metabolites. Red plot represent up-regulate, green plot mean down-regulate whereas gray represent not significant.

**Figure 2 genes-15-00828-f002:**
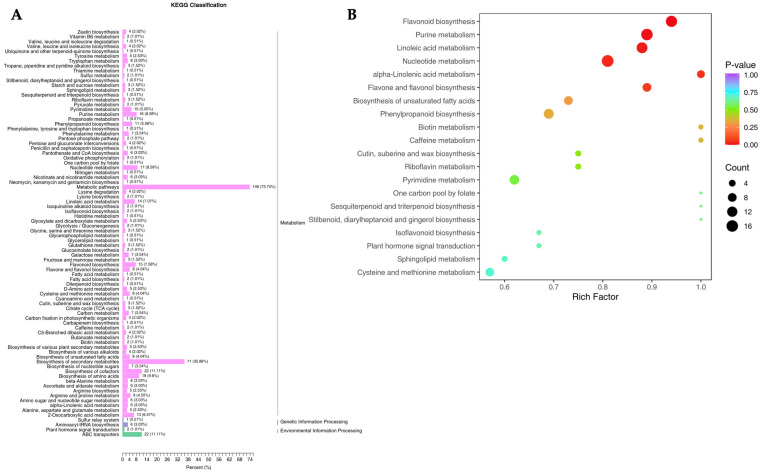
KEGG pathway enrichment of the differential metabolites in WR and RR of *P. kingianum* Coll.et Hemsl. (**A**) KEGG differential metabolite classification diagram. The ordinate is the name of the KEGG metabolic pathway, and the abscissa is the number of metabolites indicated by this pathway and its percentage associated with the total number of metabolites annotated. (**B**) Scatter diagram of the enrichment of differential metabolites in KEGG pathways.

**Figure 3 genes-15-00828-f003:**
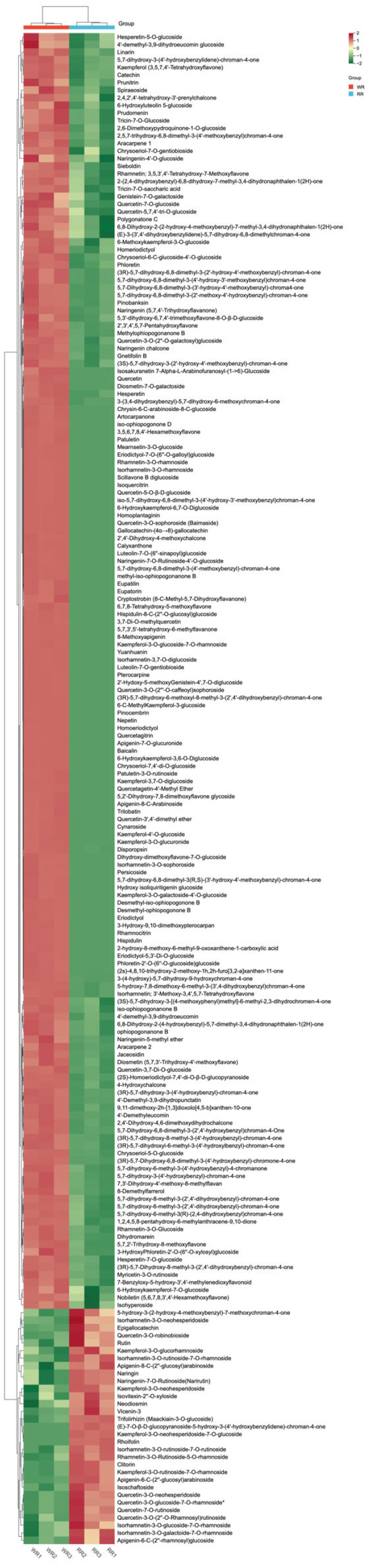
Heat maps of flavonoid metabolites biosynthesis in WR and RR of *P. kingianum* Coll.et Hemsl.

**Figure 4 genes-15-00828-f004:**
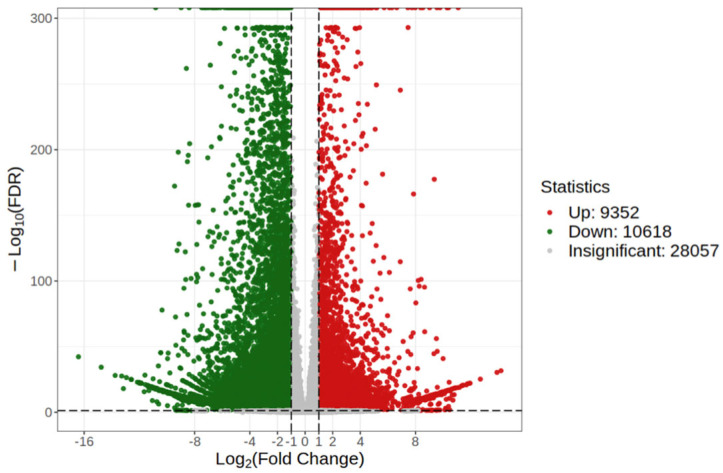
Volcano map of DEGs identified from WR and RR of *P. kingianum* Coll.et Hemsl. The abscissa is the variation in gene expression (log2FC), and the ordinate is the significant level of differentially expressed genes (-log10FDR). The green dots represent the DEGs was down-regulated, and the red dots represent the DEGs was up-regulated, and the gray dots meant the DEGs was insignificantly regulated.

**Figure 5 genes-15-00828-f005:**
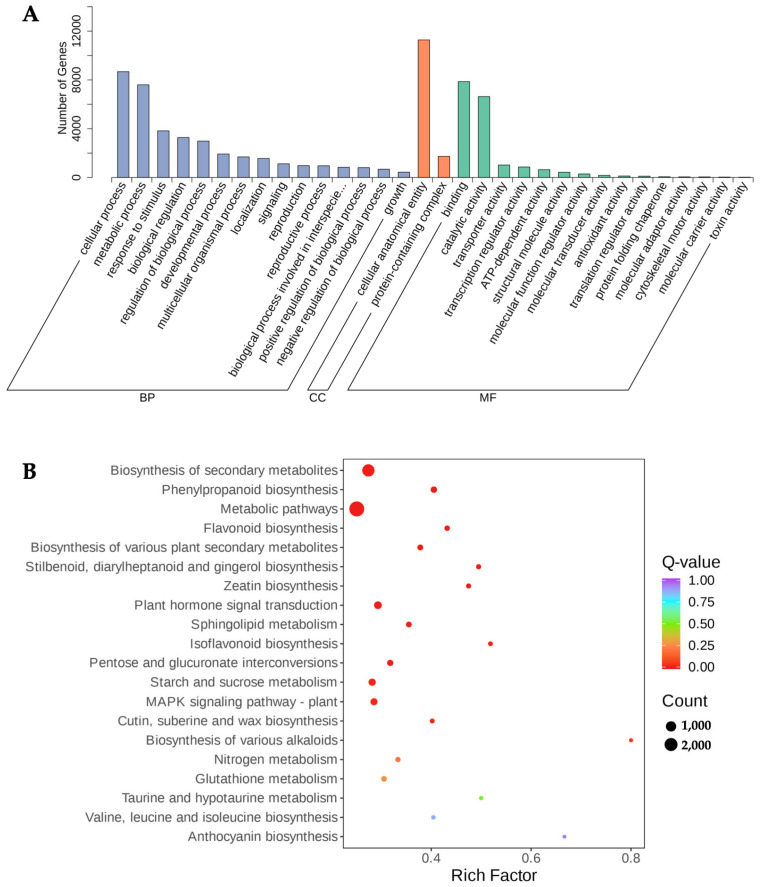
(**A**) GO and KEGG annotation of DEGs identified from WR and RR of *P. kingianum* Coll.et Hemsl. GO annotation of DEGs in WR and RR. (**B**) KEGG annotation of DEGs in WR and RR.

**Figure 6 genes-15-00828-f006:**
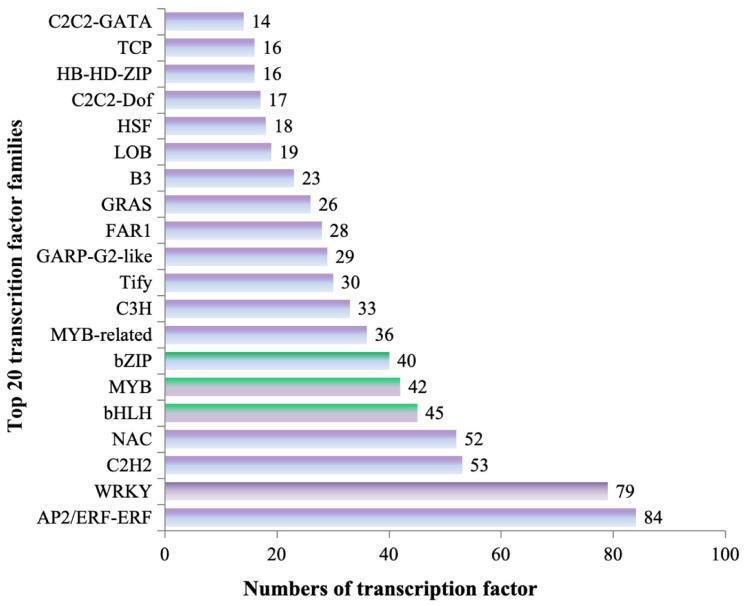
Top 20 transcript factor families (TFs) identified from WR and RR of *P. kingianum* Coll.et Hemsl. The transcription factor families involved in regulation of the biosynthesis of flavonoids was labeled with green color.

**Figure 7 genes-15-00828-f007:**
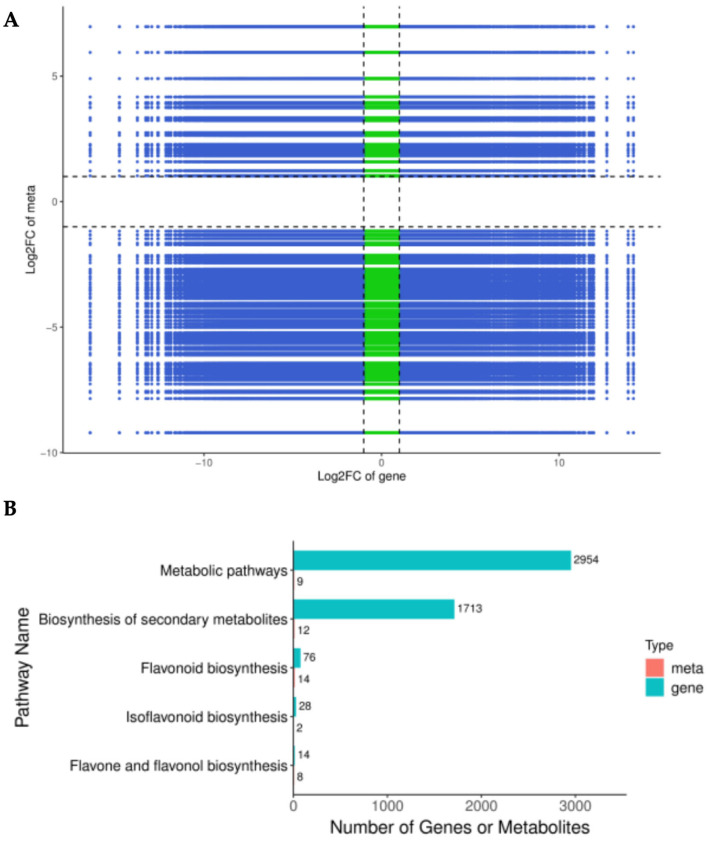
Correlation analysis between transcriptome and metabolome in WR vs. RR. (**A**) The genes correlation and components between WR and RR are shown by the nine-quadrant diagram. The abscissa represents the log2FC of the genes, and the ordinate represents the log2FC of the metabolites. (**B**) KEGG enrichment analysis of DEGs (blue column) and DAMs (red column) enriched in the same pathway. The red color represent the metabolites and the blue color represent the genes.

**Figure 8 genes-15-00828-f008:**
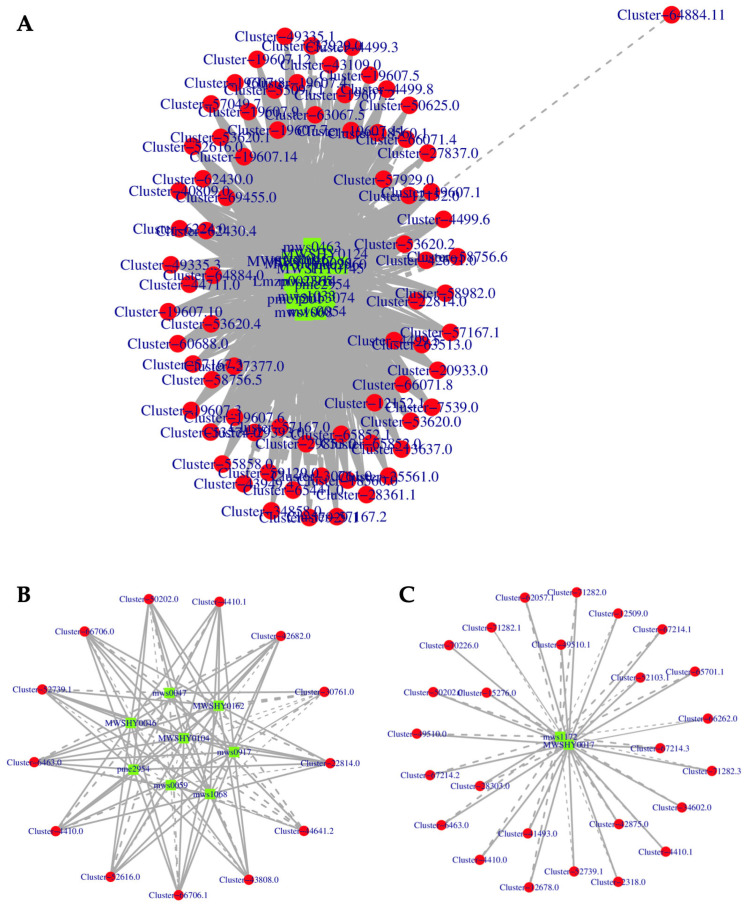
Correlation network analysis of flavonoid derivatives biosynthesis in WR and RR of *P. kingianum* Coll.et Hemsl. (**A**) Correlation network of DEGs and DAMs in flavonoid biosynthesis pathway. (**B**) Correlation network of DEGs and DAMs in isoflavonoid biosynthesis. (**C**) Correlation network of DEGs and DAMs in flavone and flavonol biosynthesis. Green and red ovals represent metabolites and genes, respectively. Solid line represents positive correlation, and dash line represents negative correlation.

**Figure 9 genes-15-00828-f009:**
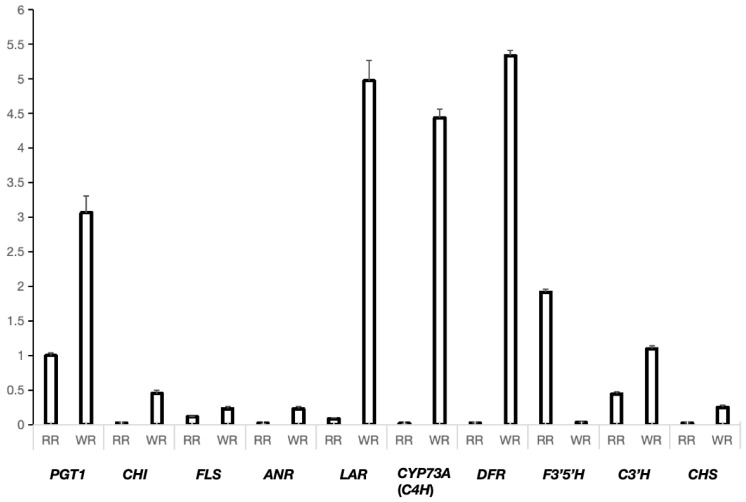
Validation of 10 DEGs by qRT-PCR in WR and RR. The X-axis mean genes in WR and RR. The left Y-axis represents the relative expression of each gene in WR and RR by qRT-PCR.

**Table 1 genes-15-00828-t001:** Primer sequence information of the genes validated by qRT-PCR.

Gene Name	Forward Primer Sequence (5′-3′)	Reverse Primer Sequence (5′-3′)
*ubiquitin*	GGACCCAGAAGTACGCAATG	AATTACCAGGGATACAGCACC
*CYP73A*(*C4H*)	GTGAACCACCCAGAAATCCAAC	GTTCTTCCAGTGGGCAGGGTTG
*C3′H*	GCCCATTAGGGAGGATGAGGTG	CACGCCCGATAGGTGATTTCTC
*CHI*	GCCTTTACTGCCATCGGAGTCTATC	GCCACGCAGTTCTCGGACACCTT
*PGT1*	GCGATGCCGACACTGAAGGA	AAGCTTCCCATGCTGCCGAA
*FLS*	CACGGATAAGGAGCGGGTGTTGA	GTAGTCCACCCACGCCTTCTTCC
*CHS*	CGGCAACCAAAGCGATAAAGGA	CCAGGTTCTGTATGTGGGTTTCG
*ANR*	GCCGAACAGTGGCATTGACAGCA	GCTTGCGTTACGGGTGGAAATGG
*DFR*	CCTTCTCCCTCGCCCTCGAC	AGAGCAGCTCCGCGTCCTTG
*F3′5′H*	GCAAGGTGATCGGCAAGGAAAGC	TCTGAACTGAACCCTTCTCGATGT
*LAR*	AGCTGCTGGAACTGTCAAGAGGT	GCCTCAGTGGCACGACGGAT

**Table 2 genes-15-00828-t002:** The top 40 significantly differential metabolites between the WR and RR of *P. kingianum* Coll.et Hemsl.

Compounds	Formula	Class	VIP	FC Value	Log2FC
Puqienine B	C_28_H_45_NO_3_	Alkaloids	1.08	0.00	−9.21
1,2,3,4-tetrahydronorharman-l-one	C_11_H_10_N_2_O	Alkaloids	1.08	0.00	−9.20
Dimethylfraxetin; 6,7,8-Trimethoxycoumarin	C_12_H_12_O_5_	Lignans and coumarins	1.08	0.00	−8.74
Benzamide	C_7_H_7_NO	Alkaloids	1.08	0.00	−8.59
5,6,7-Trimethoxycoumarin	C_12_H_12_O_5_	Lignans and coumarins	1.08	0.00	−8.49
L-Leucyl-L-Leucine	C_12_H_24_N_2_O_3_	Amino acids and derivatives	1.08	0.00	−8.37
4-*O*-Glucosyl-4-hydroxybenzoic acid	C_13_H_16_O_8_	Phenolic acids	1.08	0.00	−8.08
Ile-Phe	C_15_H_22_N_2_O_3_	Amino acids and derivatives	1.09	0.00	−8.00
Salicylic acid-2-*O*-glucoside	C_13_H_16_O_8_	Phenolic acids	1.08	0.00	−7.88
4-Hydroxychalcone	C_15_H_12_O_2_	Flavonoids	1.08	0.00	−7.85
1,5-*O*-dicaffeoyl-3-*O*-glucoside-quinic acid	C_31_H_34_O_17_	Phenolic acids	1.08	0.00	−7.80
Isochlorogenic acid A	C_25_H_24_O_12_	Phenolic acids	1.08	0.00	−7.80
1-(4′-Hydroxy-3′-methoxyphenyl)-2-[4″-(3-hydroxypropyl)-2″,6″-dimethoxyphenyl]-propane-1,3-Diol	C_21_H_28_O_8_	Others	1.08	0.00	−7.75
13-Hydroperoxy-9Z,11E-octadecadienoic acid	C_18_H_32_O_4_	Lipids	1.08	0.00	−7.73
PI(18:2/0:0)	C_27_H_49_O_12_P	Lipids	1.09	0.00	−7.71
L-Valyl-L-Leucine	C_11_H_22_N_2_O_3_	Amino acids and derivatives	1.08	0.00	−7.69
L-Valyl-L-Phenylalanine	C_14_H_20_N_2_O_3_	Amino acids and derivatives	1.08	0.01	−7.64
14-Hydroxy-diosgenin-3-*O*-xylosyl(1→4)glucoside (Ophiopogonin S)	C_38_H_60_O_13_	Steroids	1.08	0.01	−7.61
4′,5,7-Trihydroxy-3′,6-dimethoxyflavone (Jaceosidin)	C_17_H_14_O_7_	Flavonoids	1.08	0.01	−7.61
LysoPC 18:1(2n isomer)	C_26_H_52_NO_7_P	Lipids	1.09	0.01	−7.60
3,22,26-Trihydroxy-furost-5-en-12-one-3-*O*-glucosyl(1→4)fucoside-26-*O*-glucoside (Kingianoside D)	C_45_H_72_O_19_	Steroids	1.04	14.60	3.87
Clitorin	C_33_H_40_O_19_	Flavonoids	1.07	14.92	3.90
Kaempferol-3-*O*-rutinoside-7-*O*-rhamnoside	C_33_H_40_O_19_	Flavonoids	1.07	14.92	3.90
Trigonelline	C_7_H_7_NO_2_	Alkaloids	1.08	15.40	3.94
Quercetin-7-*O*-rutinoside	C_27_H_30_O_16_	Flavonoids	1.06	15.42	3.95
3-Carbamyl-1-methylpyridinium;(1-Methylnicotinamide)	C_7_H_9_N_2_O	Alkaloids	1.09	15.64	3.97
Yamogenin-Glc-Glc-Rha-Xyl	C_50_H_80_O_21_	Steroids	1.08	16.24	4.02
Octyl 6-*O*-α-L-Arabinopyranosyl-β-D-Glucopyranoside	C_19_H_36_O_10_	Others	1.08	16.72	4.06
Yamogenin-3-*O*-Neohesperidoside	C_39_H_62_O_12_	Steroids	1.07	17.07	4.09
Trifolirhizin (Maackiain-3-*O*-glucoside)	C_22_H_22_O_10_	Flavonoids	1.07	17.99	4.17
Diosgenin-3-*O*-rhamnosyl(1→2)[glucosyl(1→3)]glucoside (Gracillin)	C_45_H_72_O_17_	Steroids	1.08	18.31	4.19
Yamogenin-Glc-Glc-Rha	C_45_H_72_O_17_	Steroids	1.08	18.31	4.19
Diosgenin-Glc-Glc-Rha-Xyl	C_50_H_80_O_21_	Steroids	1.00	20.24	4.34
1-Octen-3-Ol-3-*O*-β-D-Xylopyranosyl(1→6)-β-D-Glucopyranoside	C_19_H_34_O_10_	Others	1.07	24.77	4.63
Sarsasapogenin-3-*O*-glucosyl(1→2)[rhamnosyl(1→4)]glucoside (Asparanin B)	C_45_H_74_O_17_	Steroids	1.09	25.58	4.68
Quercetin-3-*O*-(2″-*O*-Rhamnosyl)rutinoside	C_33_H_40_O_20_	Flavonoids	1.04	29.91	4.90
5(6)-En-spirost-3-ol-3-*O*-glucosyl-(1→4)-[rhamnosyl-(1→2)]-galactoside	C_45_H_72_O_17_	Steroids	1.08	44.10	5.46
Isorhamnetin-3-*O*-rutinoside-7-*O*-rutinoside	C_40_H_52_O_25_	Flavonoids	1.07	61.60	5.94
N-α-Acetyl-L-ornithine	C_7_H_14_N_2_O_3_	Amino acids and derivatives	1.08	105.46	6.72
N-α-Acetyl-L-Asparagine	C_6_H_10_N_2_O_4_	Amino acids and derivatives	1.08	125.35	6.97

Note: VIP, the variable importance in projection. FC, the fold change.

**Table 3 genes-15-00828-t003:** Unigenes involved in the flavonoid biosynthetic pathway of *P. kingianum* Coll.et Hemsl.

Number	Gene Abbreviations	Name	Unigene Quantity	EC Number	Pathway	Source
1	*HCT*	Shikimate *O*-hydroxycinnamoyltransferase	16	2.3.1.133	Ko00940 Ko00941	*P. kingianum* [37] *P. cyrtonema* [38]
2	*CCOMT*	Caffeoyl-CoA *O*-methyltransferase	7	2.1.1.104	Ko00940 Ko00941	*P. kingianum* [37,39] *P. cyrtonema* [38]
3	*CYP73A*(*C4H*)	Trans-cinnamate 4-monooxygenase	4	1.14.14.91	Ko00940 Ko00941	*P. kingianum* [37,39] *P. cyrtonema* [38]
4	*C3′H*	5-*O*-(4-coumaroyl)-d-quinate 3′-monooxygenase	1	1.14.14.96	Ko00940 Ko00941	*P. kingianum* [39] *P. cyrtonema* [38]
5	*CHI*	Chalcone isomerase	10	5.5.1.6	Ko00941	*P. kingianum* [37,39] *P. cyrtonema* [38]
6	*PGT1*	Phlorizin synthase	11	2.4.1.357	Ko00941	
7	*FLS*	Flavonol synthase	3	1.14.20.6	Ko00941	*P. kingianum* [39] *P. cyrtonema* [38]
8	*F3′H*	Flavonoid 3′-monooxygenase	2	1.14.14.82	Ko00941 Ko00944	*P. kingianum* [39] *P. cyrtonema* [38]
9	*CHS*	Chalcone synthase	15	2.3.1.74	Ko00941	*P. kingianum* [37,39] *P. cyrtonema* [38]
10	*ANR*	Anthocyanidin reductase	1	1.3.1.77	Ko00941	*P. kingianum* [39]
11	*DFR*	Bifunctional dihydroflavonol 4-reductase	4	1.1.1.219	Ko00941	*P. kingianum* [39] *P. cyrtonema* [38]
12	*F3* *′* *5* *′* *H*	Flavonoid 3′,5′-hydroxylase	1	1.14.14.81	Ko00941	safflower [40]
13	*LAR*	Leucoanthocyanidin reductase	1	1.17.1.3	Ko00941	*P. kingianum* [39] *P. cyrtonema* [38]

## Data Availability

The data presented in this study are available on request from the corresponding author. The data are not publicly available due to privacy.

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
