# Peer review of "Combined Metabolomics and Transcriptomics Analysis of the Distribution of Flavonoids in the Fibrous Root and Taproot of Polygonatum kingianum Coll.et Hemsl"

_genes, 2024, doi:10.3390/genes15070828_

Round 1

Reviewer 1 Report

Comments and Suggestions for Authors

Dear Authors,

The manuscript shows interesting findings associated with flavonoid biosynthesis by combining the transcriptome and metabolomic analysis, however, major considerations need to be made along the text and also, additional analysis (mentioned in the suggestions).    

Major:

- Section 2.2, Plant Materials and Treatments: There were selected only one plant per region to collect all samples. Why were not collected samples from different plants as biological replications in each region?  

- Lines 242, 243, and 244 are confusing, first say that have 170 flavonoids for both fibrous root and taproot then after say that have 153 flavonoids in taproot. Please rewrite this part.

- Supplementary table S7, it is unclear what is described in lines 289, 290, and 291. Also, the meaning of the highlighted row.  

- Check carefully the citations, some of the citations are related to other Polygonatum spp. then P. kingianum Coll.et Hemsl. Change for others related to P. kingianum or others that are related to Polygonatum spp.

- Need a validation analysis for DEGs (qRT-PCR), chose some of the main DEGs.

- The discussion section can better explore the network between the DEGs and the DAMs, point specific genes, and how they are contributing to the biosynthesis of the metabolites found.

- Along the text is not clear if have or no difference between regions.

- All figures' legends need to be improved, add a better description of the content in the figure.

- The text shows a lot of misspellings.

Minor:

- Citation 3 in line 39 refers only to Polygonatum sibiricum, not Polygonatum kingianum.

- Line 77: correct ‘Metarials’ and ‘ang’

- Section 2.2 ‘Plant Materials and Treatments is not clear how many biological replications.

- Lines 83 and 84 do not need to write the full names of WR and RR again, it was just mentioned in two lines above.

- Lines 83 to 85 are confusing and need to be rewritten. You already mentioned that WR and RR samples were collected for analysis. Also, in line 85, the sentence ‘3 samples analyzed for each …’ sounds better ‘3 samples collected for each …’

- Line 87, was already mentioned in the previous topic.

- Line 95, correct ‘Date’

- Line 96. Write the full name for the abbreviations UPLC-ESI-MS/MS.

- In the Material and Methods section, please provide citation for each protocol.  

- Section 2.4 ‘ Transcriptomic Analysis and Data Processing’ line 134: How the RNA was extracted? How cDNA was synthesized, and how the DEGs were obtained, which tool was used?

- Result section: Use the abbreviations WR and RR as shown in the figures.

- Figure 1: Poor resolution and needs font size increase (A and C). Figure letters (A, B, C, and D) must follow the same standard (If in the bottom, all need to be. I suggest put in the top of the figure). Figure legend, put the distinguishing letters as shown in the figure.

- Line 188: Indicate the Figure letter in ‘(Figure 1)’.

- Table 1: Show the full name of the VIP in a table description at the bottom.

- Figure 2: Unreadable, poor resolution, needs font size increase, and as the other figures, need figure letters standardization.

- Figure 3: Figure legend needs more figure content description, poor resolution, and needs font size increase (unreadable).

- Line 261, correct ‘EDGs’

- Figure 4 does not show much information, it can be moved to supplemental data or group with figures 5 and 6.

- Figure 4, needs to rewrite the legend in a better way, especially how the dots’ colors represent the DEGs.

- Line 314, change Figure 6A to 8A.

- Line 320, change figure number.

- Figure 9, change the figure letters' order to start with A, B, and C in the text.

- Line 351, citation 27 does not correspond to the sentence. 

Comments on the Quality of English Language

Constant misspellings can be identified in the text. The text needs an English revision. 

Reviewer 2 Report

Comments and Suggestions for Authors

The combined metabolomics and transcriptomics analysis of the distribution of flavonoids in the fibrous and taproot of Polygonatum has been conducted diligently. The interact-omics data generated is a very good addition to the understanding of the plant’s roots and in its use in medical applications.

The document is well written, with exceptions of some sentences not written in the correct tense. The document also needs some formatting for spacing and minor spell checks. Some of the other corrections are mentioned below.

Line 11: Correct the mentions of Polygonati Rhizoma to Polygonatum rhizome.

Line 48: Solanaceae [14]. Mention which plants in this family were studied in this reference.

Line 74: Italicize scientific names. Check in the document for any such formatting.

Line 77: Plant materials and treatments

Lines 90, 150, 152, : Correct the tense of these sentences. Check the document for other such corrections.

Line 99: The mobile phase consisted of …

Line 181: performed, evaluate – to correct

Table 1: Top 40 significantly

Figure 2: The font size and resolution should be increased.

Figure 3: The words are not so clear.

Supplementary tables are detailed and compiled well, but have some columns in another language. Kindly correct these. They should be mentioned in the text in the results section and associated with the figures where necessary so that the readers can have access to the specific files.

Line 264: biosynthesis

Comments on the Quality of English Language

Minor edits in sentence construction and spell check required.

Round 2

Reviewer 1 Report

Comments and Suggestions for Authors

Dear authors,

Thank you for considering the suggested revisions.

The text was well corrected, however, most figures (1, 2, 3, 5, 7, and 8) still present low-resolution quality. The figure's text is unreadable.
